# Suboptimal Concentrations of Ceftazidime/Avibactam (CAZ-AVI) May Select for CAZ-AVI Resistance in Extensively Drug-Resistant *Pseudomonas aeruginosa*: In Vivo and In Vitro Evidence

**DOI:** 10.3390/antibiotics11111456

**Published:** 2022-10-22

**Authors:** Inmaculada Lopez-Montesinos, María Milagro Montero, Sandra Domene-Ochoa, Carla López-Causapé, Daniel Echeverria, Luisa Sorlí, Nuria Campillo, Sonia Luque, Eduardo Padilla, Nuria Prim, Santiago Grau, Antonio Oliver, Juan P. Horcajada

**Affiliations:** 1Infectious Diseases Service, Hospital del Mar, 08003 Barcelona, Spain; 2Infectious Pathology and Antimicrobials Research Group (IPAR), Institut Hospital del Mar d’Investigacions Mèdiques (IMIM), 08003 Barcelona, Spain; 3Department of Medicine, Universitat Autònoma de Barcelona (UAB), 08193 Bellaterra, Spain; 4Department of Medicine and Life Sciences (MELIS), Universitat Pompeu Fabra Barcelona, 08002 Barcelona, Spain; 5CIBER of Infectious Diseases (CIBERINFEC CB21/13/00002 and CB21/13/00099), Institute of Health Carlos III, 28029 Madrid, Spain; 6Servicio de Microbiología y Unidad de Investigación, Hospital Son Espases, IdISBa, 07120 Palma de Mallorca, Spain; 7Pharmacy Service, Hospital del Mar, 08003 Barcelona, Spain; 8Microbiology Service, Laboratori de Referència de Catalunya, 08820 Barcelona, Spain

**Keywords:** ceftazidime/avibactam, *Pseudomonas aeruginosa*, multidrug-resistant, continuous infusion, PK/PD, hollow fiber

## Abstract

This study correlates in vivo findings in a patient with an extensively drug-resistant (XDR) *P. aeruginosa* infection who developed resistance to ceftazidime-avibactam (CAZ-AVI) with in vitro results of a 7-day hollow-fiber infection model (HFIM) testing the same bacterial strain. The patient was critically ill with ventilator-associated pneumonia caused by XDR *P. aeruginosa* ST175 with CAZ-AVI MIC of 6 mg/L and was treated with CAZ-AVI in continuous infusion at doses adjusted for renal function. Plasma concentrations of CAZ-AVI were analyzed on days 3, 7, and 10. In the HIFM, the efficacy of different steady-state concentrations (Css) of CAZ-AVI (12, 18, 30 and 48 mg/L) was evaluated. In both models, a correlation was observed between the decreasing plasma levels of CAZ-AVI and the emergence of resistance. In the HIFM, a Css of 30 and 48 mg/L (corresponding to 5× and 8× MIC) had a bactericidal effect without selecting resistant mutants, whereas a Css of 12 and 18 mg/L (corresponding to 2× and 3× MIC) failed to prevent the emergence of resistance. CAZ/AVI resistance development was caused by the selection of a single ampC mutation in both patient and HFIM. Until further data are available, strategies to achieve plasma CAZ-AVI levels at least 4× MIC could be of interest, particularly in severe and high-inoculum infections caused by XDR *P. aeruginosa* with high CAZ-AVI MICs.

## 1. Introduction

The emergence and spread of extensively drug-resistant (XDR) *Pseudomonas aeruginosa* has become a matter of public health concern. The increase in XDR strains seriously compromises antibiotic treatment options [1]. Receiving ineffective antibiotic therapy is associated with worse outcomes and higher mortality rates among patients with *P. aeruginosa* infections [2,3].

In recent years, the availability of new drugs such as ceftazidime-avibactam (CAZ-AVI) has increased the therapeutic arsenal against these microorganisms [4]. However, the emergence of resistant mutants has already been reported [5] and strategies to monitor and prevent the selection of resistance during antibiotic treatment are urgently needed.

CAZ-AVI (Zavicefta^®^) [6] is a ceftazidime/β-lactamase inhibitor combination with activity against extended-spectrum beta-lactamase (ESBL) producers and carbapenem-resistant Enterobacterales (CRE), but is not active against metallo-β-lactamase (MBL) producers. The addition of avibactam to ceftazidime protects the cephalosporin from enzymatic degradation caused by *P. aeruginosa* strains (mainly due to Amp-C enzymes but also ESBLs and class A carbapenemases) and leads to decreased minimum inhibitory concentrations (MICs) of ceftazidime [7]. It has been approved by the U.S. Food and Drug Administration and the European Medicines Agency for the treatment of infections caused by *P. aeruginosa* strains from different sources (i.e., hospital-acquired bacterial pneumonia, complicated intra-abdominal infections, and urinary tract infections). 

The current recommended dosage of CAZ-AVI for adult patients with normal renal function is a 2 h infusion of 2 g/0.5 g every 8 h. As a time-dependent antibiotic, the best pharmacokinetic parameter to achieve maximum bacterial killing is the percentage of free drug concentration that remains above the MIC (%ƒT > MIC) for 40–70% [8] of the dosing interval, and is maximized when concentrations in plasma are 4–5× MIC.

The recommended standard dosing regimen may be insufficient to treat infections caused by isolates with CAZ-AVI MIC values close to susceptibility breakpoints of 8 mg/L, due to the increased likelihood of not achieving effective concentrations [9]. In line with this, it has been suggested that the therapeutic target could be increased to 100% ƒT > 4–8 times MIC of the free drug [10,11] in complicated scenarios, such as critically ill patients or high inoculum infections such as pneumonia. 

The use of prolonged infusion could offer advantages for attaining pharmacokinetic/pharmacodynamic (PK/PD) targets and optimizing antibiotic treatment [12]. In a randomized clinical trial including 60 patients with severe sepsis treated with b-lactam therapy, 82% and 29% of those receiving continuous infusion (CI) and intermittent dosing, respectively, achieved 100% ƒT ≥ MIC against target pathogens. In addition, the clinical cure rate was higher in the CI group than in the intermittent administration group (70% versus 43%; *p* = 0.037). In the specific case of CAZ-AVI, Goncette et al. [10] assessed its performance when administered by CI in a case series of 10 patients. The median daily dose of CAZ-AVI used was 10 g (interquartile range 5–10 g) with median reported steady-state concentrations (Css) level of 63.6 mg/L (interquartile range 47.6–80 mg/L). In this setting, CAZ-AVI in CI achieved clinical cure and microbiological eradication rates of 80% and 90%, respectively. 

This study is based on an actual clinical experience in which a critically ill patient with pneumonia caused by an XDR *P. aeruginosa* strain with ceftazidime-avibactam MIC of 6 mg/L developed resistance to this agent. The patient was included in the PseudoNOVA observational study (see Methods section).

We hypothesized that the administration of higher doses of CAZ-AVI in CI could help optimize the PK/PD target for CAZ-AVI and prevent the emergence of resistant mutants in isolates with borderline MICs. To test this, we used an in vitro hollow-fiber infection model (HFIM) to evaluate the efficacy of three dosing regimens of CAZ-AVI given by CI against an XDR *P. aeruginosa* strain with MIC of 6 mg/L isolated from the mentioned patient and correlated these findings with the in vivo results. We also characterized emerging resistance mechanisms by whole genome sequencing (WGS). 

## 2. Results

### 2.1. Clinical Study

We present the 2018 case of a 55-year-old female patient with a prior history of obesity (weight 90 kg, height 160 cm: body mass index 35 kg/m^2^), type 2 diabetes mellitus and chronic kidney disease in need of a kidney transplant, who was receiving long-term immunosuppressive treatment with everolimus 1 mg every 24 h, mycophenolate 360 mg every 8 h, and prednisone 5 mg every 24 h. She was included in the PseudoNOVA study due to ventilator-associated pneumonia caused by an XDR *P. aeruginosa* strain recovered from a bronchial aspirate (BAS) sample, which was susceptible to CAZ-AVI with MIC of 6 mg/L. A 15-day course of directed therapy with CAZ-AVI was prescribed, consisting of a loading dose of 2 g/0.5 g followed by 3 g/0.75 g given in CI every 24 h, adjusted for renal function (median Glomerular filtration rate (GFR), 51.5 mL/min). Plasma CAZ-AVI levels on days 3 and 7 were 81.4 mg/L and 77.4 mg/L, respectively. A control BAS confirmed no growth of *P. aeruginosa*. When the patient was assessed at the end of the treatment window, clinical cure and microbiological eradication were considered to have been achieved and she was extubated. 

However, 20 days after the index episode, the patient developed acute tracheobronchitis. A new BAS was performed and XDR *P. aeruginosa* was again isolated. The CAZ-AVI MIC was 6 mg/L. In addition, CAZ-AVI-resistant subpopulations with MIC of 32 mg/L were detected in the same culture. CAZ-AVI was reinitiated at the same dosage, achieving plasma levels of CAZ-AVI of 54 mg/L, 58.1 mg/L and 27 mg/L on days 3, 7 and 10, respectively (median GFR of 91.5 mL/min). The patient showed a good clinical response and CAZ-AVI was stopped after 10 days of antibiotic treatment. Two further follow-up BAS were performed 1 day and 10 days after the end of antibiotic treatment. CAZ-AVI-resistant subpopulations of XDR *P. aeruginosa* were documented in both samples, with CAZ-AVI MICs of 24 to 48 mg/L, and 32 to 256 mg/L, respectively. They were interpreted as colonization. The patient was assessed as clinically cured, but with microbiological failure at the follow-up endpoint. 

In both episodes (ventilator-associated pneumonia and acute tracheobronchitis), the patient received adjuvant treatment with nebulized colistimethate sodium (CMS) at doses of 2 million international units every 8 h. 

### 2.2. HFIM

The initial *P. aeruginosa* strain was exposed to different concentrations of CAZ-AVI administered as CI in a 7-day in vitro HFIM. Total colony-forming unit (CFU)/mL reductions observed for the different regimens of CAZ-AVI during the HFIM are shown in Figure 1. The administration of CAZ-AVI in CI was bactericidal at concentrations of 5× and 8× MIC, corresponding to Css of 30 mg/L and 48 mg/L, but not at lower concentrations (Css of 12 mg/L and 18 mg/L). The mean bacterial density of the starting inoculum was 7.08 log_10_ CFU/mL. The behavior of CAZ-AVI in CI at Css of 12 mg/L and 18 mg/L was similar to the control regimen without antibiotic, with final bacterial densities of 8.65 and 8.18 log_10_ CFU/mL, respectively. CAZ-AVI in CI at higher concentrations (Css of 30 mg/L and 48 mg/L) achieved a continuous bacterial reduction of 4.16 and 4.48 log_10_ CFU/mL with bacterial densities of 2.92 and 2.60 log_10_ CFU/mL, respectively, at the end of the experiment.

### 2.3. Resistance Studies

In the 7-day HFIM study, CAZ-AVI-resistant mutants were not selected when CAZ-AVI was administered in CI at Css of 30 mg/L and 48 mg/L. Nevertheless, CAZ-AVI-resistant subpopulations emerged at lower concentrations of CAZ-AVI, corresponding to Css of 12 mg/L and 18 mg/L, administered in CI (Figure 2 and Appendix A).

### 2.4. In Vitro Susceptibility and Resistance Mechanisms

MLST analysis from WGS revealed that the XDR clinical isolates studied belonged to the widespread ST175 high-risk clone. The initial isolate was susceptible to ceftazidime-avibactam (MIC 6 mg/L) and resistant to the classical β-lactams (piperacillin/tazobactam, ceftazidime, cefepime, aztreonam, imipenem and meropenem) due to an inactivating mutation in OprD (Q142X) and AmpC hyperproduction (G154R mutation in AmpR), as described previously for ST175 isolates. The subsequent CAZ-AVI-resistant clinical isolates differed from the parent strain by a single SNP that resulted in the previously described Q146K mutation in AmpC [13]. The CAZ-AVI-resistant mutants obtained from the HFIM similarly differed from the parent strain by a single mutation in *ampC*, in this case a 21 bp deletion leading to the deletion of 7 amino acids (positions 236 to 242) in the Ω-loop (Table 1). 

### 2.5. Drug Concentrations

The relation between observed and predicted CAZ-AVI concentrations over 7 days is shown in Appendix A. Agreement between observed and predicted results was evaluated with a Bland–Altman plot. For all Css of 12 and 18 mg/L, difference values lay within 1.96 standard deviations (SD) of the mean. For Css of 30 and 48 mg/L, on the other hand, two and three of the values deviated slightly from ±1.96 SD in 18 and 30 Css, respectively.

## 3. Discussion

In this translational study, the clinical experience of an actual patient was correlated with in vitro HFIM findings. We evaluated a critically ill patient with relevant comorbidities and ventilator-associated pneumonia and tracheobronchitis due to XDR *P. aeruginosa* ST175 with a borderline CAZ-AVI MIC at baseline, who developed resistance to CAZ-AVI during treatment. Using HFIM, we compared three concentrations of CAZ-AVI in CI against the index isolate to identify the most efficient way to administer the antibiotic and prevent the emergence of resistance in future cases. 

In the clinical setting, we observed a correlation between decreasing levels of CAZ-AVI in plasma and the emergence of resistance to this drug. In the ventilator-associated pneumonia episode, CAZ-AVI plasma concentrations of T > 4–8 times the MIC (plasma levels of 81.4 mg/L and 77.4 mg/L on days 3 and 7) were achieved. The patient was cured, and a follow-up respiratory sample confirmed eradication of *P. aeruginosa.* With respect to the tracheobronchitis episode, selection of resistant mutants became apparent mainly as a result of the sharp drop in CAZ-AVI in plasma during the study (plasma levels of 54 mg/L, 58.1 mg/L and 27 mg/L on days 3,7 and 10, respectively). 

Several factors may have led to the reduction in CAZ-AVI concentrations. First, CAZ-AVI is mainly eliminated by the renal route [6]. In the present clinical case, an improvement in renal clearance was observed (median GFR rose from 51 mL/min to 91.5 mL/min in the pneumonia and tracheobronchitis episodes, respectively). In the second episode, however, CAZ-AVI doses were not adjusted for renal function. This most likely favored higher CAZ-AVI elimination rates and consequently lower plasma levels. It should be noted that increased renal clearance is frequently seen in critically ill patients with normal serum creatinine concentrations [14]. Furthermore, the PK of hydrophilic antimicrobials such as beta-lactams is affected by the presence of sepsis, leading to a potential increase in the volume of distribution [14]. Finally, obese patients may also have an increased volume of distribution and higher renal clearance, resulting in antibiotic exposure that is difficult to predict [11]. 

Of note, CAZ-AVI-resistant subpopulations with MIC of 32 mg/L were observed in the first respiratory sample from the tracheobronchitis episode, despite the fact that no microorganism growth had been documented in an earlier sample. Selection of CAZ-AVI-resistant subpopulations could be the consequence of lower CAZ-AVI concentrations at the site of infection. It has previously been reported that the ratio of epithelial lining fluid exposure to concentrations of ceftazidime and avibactam is about 30% of plasma exposure [15]. To prevent subtherapeutic antibiotic concentrations, previous authors [10] have suggested CAZ-AVI concentrations of ≥4–5 times the MIC at the site of the infection as the PK/PD target, rather than in plasma. This could be an interesting strategy, especially in complicated circumstances such as critically ill patients, isolates with elevated MICs and/or deep-seated infections. However, antibiotic-related side effects should be carefully monitored.

In the present clinical case, the patient had a favorable outcome in terms of clinical cure despite developing resistance to CAZ-AVI. This may be due in part to the fact that CAZ-AVI resistance was documented during the tracheobronchitis episode, which is considered a low-risk source of infection [2]. Indeed, the need for systemic antibiotics in this type of infection is controversial [16]. 

On the other hand, the emergence of CAZ-AVI resistance may further complicate antimicrobial therapy. In addition, borderline MICs may lead to the appearance of low-level resistance mechanisms that can ultimately compromise clinical outcome. In the case of ceftolozane-tazobactam, it was reported that higher MICs (>2 mg/L) predict 30-day mortality in patients with lower respiratory tract infections caused by MDR or XDR *P. aeruginosa* [17]. 

Although there are no clinical data to support a different therapeutic management that takes MIC values into account, it is obvious that a higher MIC will reduce the likelihood of achieving any PK/PD target, including ƒT > MIC. In this scenario, strategies such as the use of higher doses of CAZ-AVI, extended therapy, CI and/or combination therapy may be considered. In the latter circumstance, our group [18] demonstrated in vitro that combinations of CAZ-AVI plus colistin, amikacin or aztreonam were additive or synergistic in at least 85% of the XDR *P. aeruginosa* isolates studied, including CAZ-AVI-resistant *P. aeruginosa*. Clinical experience has also shown that administration of CAZ-AVI by prolonged infusion (≥3 h) reduces mortality by 46% [19].

In the HFIM, our in vitro findings showed similar trends to the in vivo results. A Css of 30 and 48 mg/L (corresponding to 5× and 8× MIC) had a bactericidal effect without selecting for resistant mutants, whereas a Css of 12 and 18 mg/L (corresponding to 2× and 3× MIC) clearly failed to prevent the emergence of resistance. WGS revealed that the development of CAZ-AVI resistance in XDR *P. aeruginosa* ST175 was caused by selection of a single *ampC* mutation, both in the patient and in the HFIM. Our results agree therefore with previous findings for CAZ-AVI and ceftolozane-tazobactam that point to AmpC mutations as the major resistance mechanism [20,21]. It should be noted that resistance emerged in our case, as in previous ones, in XDR strains that were already ceftazidime-resistant due to a mutation leading to AmpC overexpression, which can be considered a prerequisite for subsequent CAZ-AVI and ceftolozane-tazobactam resistance development due to selection of AmpC structural mutations [22]. These results highlight the importance of optimizing antibiotic treatment, particularly in a rapidly adapting microorganism such as *P. aeruginosa*. 

The main limitation of the study is that only a single isolate was studied, although it belongs to an XDR *P. aeruginosa* clone that is widespread in Spanish hospitals [15]. More in vivo and in vitro examples should be analyzed before drawing generalizable conclusions. Second, we evaluated plasma concentrations of ceftazidime but not avibactam. Considering the potential nonlinear synergetic effect between CAZ and AVI, using CAZ plasma concentration alone could be controversial. Conversely, CAZ and AVI displayed similar PK/PD profile in terms of lung penetration, volume of distribution, time-dependent activity, low plasma protein binding and renal clearance in previous reports [6,15,23]. It would also have been interesting to measure CAZ and AVI concentrations in epithelial lining fluid to corroborate hypothetical subtherapeutic concentrations at the site of infection. Another limitation is potential variation due to the method of MIC determination [24]. These variations must be taken into account to prevent potential under- or overdosing of patients. Finally, due to the observational nature of the study, the CAZ-AVI dose was decided by the physicians in charge and was not therefore accurately adjusted for renal clearance in accordance with current recommendations. Nevertheless, this reflects daily clinical practice and enabled us to test our hypothesis in vivo. As a strength, this is one of the few reports in the literature to correlate clinical studies with HFIM findings and represents an example of translational research to clinics. 

## 4. Materials and Methods

### 4.1. The PseudoNOVA Project

The PseudoNOVA project is a Spanish prospective, multicenter, observational cohort study, conducted between 2018 and 2022, of the clinical and microbiological impact of the new antipseudomonal agents ceftolozane-tazobactam and CAZ-AVI on infections caused by high-risk clones of XDR *P. aeruginosa* in Spain. Correlations with in vitro results were studied using a hollow-fiber dynamic PK/PD model. Patients admitted to participating hospitals during the study period with invasive infections caused by XDR *P. aeruginosa* and treated with ceftolozane-tazobactam or CAZ-AVI were evaluated in terms of mortality, clinical cure, microbiological eradication, and selection of resistant mutants. Antibiotic regimen and dose selection were decided by the physician in charge without interference from the team of investigators. GFR was calculated using the Chronic Kidney Disease Epidemiology Collaboration (CKD-EPI). In this study, plasma levels of ceftolozane-tazobactam or CAZ-AVI were performed blinded on days 3, 7, 14 and 21 of treatment (as appropriate). Strains that developed resistance during antibiotic treatment were selected for study in a HFIM with a view to designing the most efficient strategies of antibiotic administration and to prevent the development of resistant mutants. 

### 4.2. Bacterial Isolates, Microbiological Studies, and Resistance Mechanisms

Local microbiology laboratories used standard microbiological techniques for the isolation, identification, and susceptibility testing of bacteria. *P. aeruginosa* isolates were considered XDR according to *Magiorakos* et al. [25]. CAZ-AVI MICs were determined by E-test and interpreted using the European Committee on Antimicrobial Susceptibility Testing (EUCAST) recommendations [26]. Strains of *P. aeruginosa* were frozen at −80 °C for subsequent study in the HFIM.

### 4.3. Antibiotics

CAZ-AVI (Zavicefta^®^) was provided by Pfizer (Ringaskiddy, County Cork, Ireland). CI dosing regimens of CAZ-AVI were simulated to achieve Css of 12, 18, 30 and 48 mg/L. The different Css were chosen to analyze concentrations above and below the therapeutic objective of 100% ƒT ≥ 4–5 times the MIC in plasma (2×, 3×, 5×, 8× MIC of the isolate).

### 4.4. HFIM

A 7-day, experimental HFIM was carried out in duplicate, as described previously [27]. The efficacy of different CI, Css of CAZ-AVI (12, 18, 30 and 48 mg/L) was evaluated. Polyethersulfone hemofilters were used as hollow-fiber cartridges with a volume of 50 mL (Aquamax HF03, Nikkiso, Belgium) [28]. Experiments were conducted in a humidified incubator set at 37 °C. Antibiotics were pumped directly into the central reservoir with a separate infusion pump to achieve the target Css. Treatment regimens were compared with a no-treatment control. Fresh drug-free growth medium (cation-adjusted Mueller–Hinton broth [CA-MHB]) was continuously infused into the central reservoir to dilute and simulate human drug clearance. An equal volume of drug-containing medium was removed from the central reservoir concurrently to maintain an isovolumetric system. Bacterial suspensions were inoculated into the extracapillary compartment of the hollow-fiber cartridge, where they were exposed to fluctuating drug concentrations. Bacterial samples were obtained from the cartridges at 0, 8, 24, 48, 72, 96, 144 and 168 h, then washed and suspended in solution in 1 mL Eppendorf tubes to minimize the carryover effect of the drug. Decimal serial dilutions were quantitatively cultured onto drug-free trypticase soy agar (BBL TSA II, Becton Dickinson) plates to determine bacterial densities (log_10_ CFU/mL). The lower limit of detection (LLOD) was 1.3 log_10_ CFU/mL. Bactericidal activity was defined as a reduction of 3 log10 CFU/mL from the initial bacterial density [29,30,31].

### 4.5. Antimicrobial Resistance Studies

A portion of the bacterial suspension was quantitatively cultured onto agar supplemented with CAZ-AVI at two, four, and eight times the reference MIC to evaluate amplification of resistant subpopulations. When growth was observed after 72 h, up to three colonies were selected to confirm reduced susceptibility to CAZ-AVI and be analyzed for changes in MIC values from baseline. Antibiotic susceptibility testing was performed according to the Clinical and Laboratory Standards Institute (CLSI) guidelines for broth microdilution using CA-MHB [32]. 

### 4.6. Characterization of Resistance Mechanisms by WGS

CAZ-AVI-susceptible and CAZ-AVI-resistant XDR *P. aeruginosa* isolates obtained from the patient, as well as CAZ-AVI-resistant isolates obtained from the HFIM were characterized by WGS, following previously established protocols [33]. Genomic DNA was obtained by using a commercially available extraction kit (High Pure PCR template preparation kit; Roche Diagnostics). Obtained paired-end reads were mapped to the *P. aeruginosa* PAO1 reference genome (GenBank accession: NC_002516.2) with Bowtie 2 v2.2.4 and pileup and raw files were obtained by using SAMtools v0.1.16 and PicardTools v1.140, using the Genome Analysis Toolkit (GATK) v3.4.46 for realignment around InDels. From the obtained raw files, SNPs were extracted if they met the following criteria: a quality score (Phred-scaled probability of the samples reads being homozygous reference) of at least 50, a root-mean-square (RMS) mapping quality of at least 25 and a coverage depth of at least 3 reads; excluding all ambiguous variants. As well, MicroInDels were extracted from the total pileup files when meeting the following criteria: a quality score of at least 500, an RMS mapping quality of at least 25 and support from at least one-fifth of the covering reads. After conversion of these filtered files to vcf, SNPs and InDels were annotated with SnpEff v4.2. Additionally, paired-end reads were de novo assembled using SPAdes v3.13.1 [34] for study the structural integrity of porin OprD, to determine the presence of horizontally acquired antimicrobial resistance determinants and to determine the Sequence Type [35]. Finally, sequence variants located within a set of 40 chromosomal genes (*gyrB, mexR, mexA, mexB, oprM, ampDh3, parS, parR, mexY, mexX, mexZ, galU, mexS, mexT, mexE, mexF, oprN, dacB, gyrA, nalD, nalC, dacC, pbpA, mpl, ampR, ampC, fusA1, ftsI, ampD, oprJ, mexD, mexC, nfxB, pmrA, pmrB, parC, parE, armZ, ampDh2*) were extracted and natural polymorphisms were filtered [33]. 

### 4.7. Data Availability

Genomic sequences have been deposited in the European Nucleotide Archive, under project number PRJEB55650.

### 4.8. Drug Concentrations in the HFIM

During the first 48 h (0, 3, 5, 7, 9, 23, 25 27, 29 and 47 h) of the study and once a day until the end of the study, antibiotic samples were collected from the peripheral compartment of the HFIM and immediately stored at −80 °C until analysis. Samples were taken for Css reporting and analyzed by HPLC [36]. The McWhinney BC et al. technique for beta-lactams was used [37]. The concentration–time profiles of the antibiotics were validated by means of a linear model using ADAPT II [38].

## 5. Conclusions

This case reflects “the perfect storm” leading to failure of antimicrobial drug therapy. Until further data are available, it seems reasonable to use more precise or even higher dosing regimens when using CAZ-AVI to treat severe and high-inoculum infections caused by XDR *P. aeruginosa* isolates with CAZ-AVI MICs close to the susceptibility breakpoint. Strategies aimed at achieving plasma levels at least 4 × MIC could be of interest to avoid subtherapeutic antibiotic exposure at the site of infection and prevent the emergence of resistant mutants, at least in this sub-group of patients. Clinical and microbiological studies are needed to assess the feasibility, effectiveness, and safety of this challenging question.

## Figures and Tables

**Figure 1 antibiotics-11-01456-f001:**
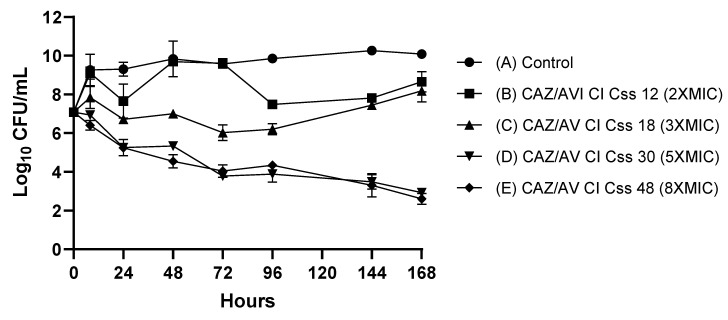
Mean reduction in bacterial density during 7-day HFIM assays (in duplicate) with the XDR *P. aeruginosa* index isolate treated with different Css of CAZ-AVI (12, 18, 30 and 48 mg/L) in CI. CFU, colony-forming unit; CI, continuous infusion; Css, steady-state concentration; MIC, minimum inhibitory concentrations. Errors bars are expressed as standard deviations (SD).

**Figure 2 antibiotics-11-01456-f002:**
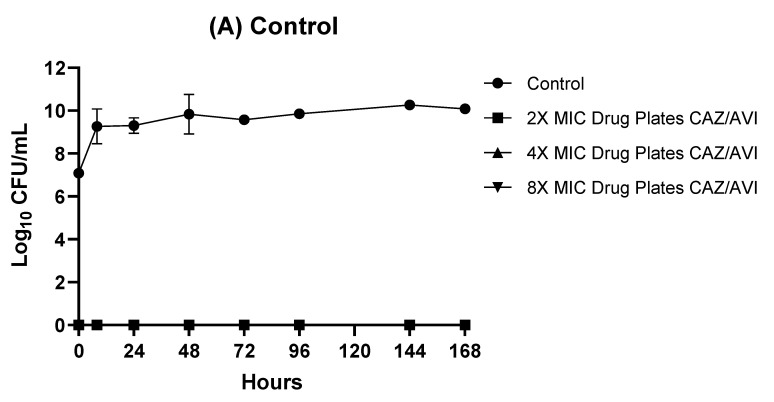
(**A**–**E**) Emergence of CAZ-AVI-resistance in the XDR *P. aeruginosa* isolate using Css of 12 (2× MIC), 18 (3× MIC), 30 (5× MIC) and 48 (8× MIC) mg/L in CI (performed in duplicate). CFU, colony-forming unit; CAZ/AVI, ceftazidime/avibactam; CI, continuous infusion; Css, steady-state concentrations; MIC, minimum inhibitory concentration. Errors bars are expressed as standard deviations (SD).

**Table 1 antibiotics-11-01456-t001:** Whole genome sequence resistome analysis of the studied *P. aeruginosa* clinical isolates and derived resistant mutants. PA, *P. aeruginosa*; CAZ/AVI, ceftazidime/avibactam; MIC, minimum inhibitory concentration; HFIM, hollow-fiber infection model.

PA Isolate	CAZ/AVI MIC	Resistome Summary
Episode 1 (Index isolate)	6 mg/L	*aadB*, *oprD* (Q142X), *mexZ* (G195D), *gyrA* (T83I, D87N), *ampR* (G154R), *parC* (L168Q, S87W), *armZ* (V266M)
Episode 2 (day 35)	6 mg/L	*aadB*, *oprD* (Q142X), *mexZ* (G195D), *gyrA* (T83I, D87N), *ampR* (G154R), *parC* (L168Q, S87W), *armZ* (V266M)
Follow up (day 46)	24 mg/L	*aadB*, *oprD* (Q142X), *mexZ* (G195D), *gyrA* (T83I, D87N), *ampR* (G154R), *parC* (L168Q, S87W), *armZ* (V266M), *ampC* (Q146K)
Follow up (day 55)	32 mg/L	*aadB*, *oprD* (Q142X), *mexZ* (G195D), *gyrA* (T83I, D87N), *ampR* (G154R), *parC* (L168Q, S87W), *armZ* (V266M), *ampC* (Q146K)
HFIM in vitro resistant mutants	>32 mg/L	*aadB*, *oprD* (Q142X), *mexZ* (G195D), *gyrA* (T83I, D87N), *ampR* (G154R), *parC* (L168Q, S87W), *armZ* (V266M), *ampC* (Δ236-242)

## Data Availability

The data presented in this study are available on request from the corresponding author. The data are not publicly available due to privacy reason.

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
