# Peer review of "Suboptimal Concentrations of Ceftazidime/Avibactam (CAZ-AVI) May Select for CAZ-AVI Resistance in Extensively Drug-Resistant Pseudomonas aeruginosa: In Vivo and In Vitro Evidence"

_antibiotics, 2022, doi:10.3390/antibiotics11111456_

Round 1

Reviewer 1 Report

The manuscript entitled “Suboptimal concentrations of Ceftazidime/Avibactam 2 (CAZ-AVI) may select for CAZ-AVI resistance in extensively drug-resistant Pseudomonas aeruginosa: In vivo and in vitro evidence.” by I Lopez Montesinos et al. reports the in vivo findings from a patient with extensively drug-resistant P. aeruginosa infection who developed resistance to CAZ-AVI. The authors further went on demonstrating in vitro results of a 7-day HFIM testing the same bacterial strain. Overall, the study is interesting and informative to dosing regimen optimization. However, several points would benefit from a more detailed analysis and better data interpretation.

1.      Though a correlation between in vivo and in vitro studies was observed, it’s not sufficient to come up with the conclusion proposed in Line 41-43. Unless the drug effect on a larger patient population with specified clinical complications (renal impairment, obese, etc.) are investigated and a comprehensive in vitro-in vivo extrapolation are developed and validated, the conclusion should be rephased with cautious.

2.      Considering the potential nonlinear synergetic effect between CAZ and AVI, using CAZ plasma concentration as a representative could be controversial. The authors should provide convincing evidence to back up their assumption. Additionally, a dose-response curve is always preferred to better characterize drug effect and further guide dose selection.

3.      A preliminary WGS analysis and supporting data (figure) should be included to demonstrate AmpC is the key mutation involved in CAZ-AVI resistance in this patient. Similarly, HFIM selected drug-resistant strain should be evaluated (verify for AmpC) and discussed to demonstrate it could serve as a qualified in vitro tool for this proof-of-concept study.

4.      To investigate the CAZ-AVI induced P. aeruginosa resistance, the development of an in vivo animal model is strongly recommended.

5.      Data interpretation

(1)    The drug concentrations presented in the graphical abstract and Fig 2 (figure legend) are different from the text, please double check. For HFIM, using 2/4/8X MIC would be better than the corresponding in vivo Css.

(2)    For Fig 1 and 2, X/Y axis should be labelled properly. A segmented X/Y axis should be considered to enable a clear view of all data points. Also, the number of replicates (n) and error bars (SD/SEM) should be included, statistical significance should be assessed.

(3)    To allow a straightforward comparison of dose-dependent resistance, a table would be preferred to sum up results from Fig 2.

(4)    A continuous timeline for the clinical treatment record (especially for the graphical abstract) will be greatly appreciated.

(5)    The data from 4 dose groups could be integrated into 1 Bland-Altman plot in supplemental figure.

(6)    The HPLC method should be included in “Materials and Methods”.

(7)    The cited references should be formatted.

Author Response

The manuscript entitled “Suboptimal concentrations of Ceftazidime/Avibactam 2 (CAZ-AVI) may select for CAZ-AVI resistance in extensively drug-resistant Pseudomonas aeruginosa: In vivo and in vitro evidence.” by I Lopez Montesinos et al. reports the in vivo findings from a patient with extensively drug-resistant P. aeruginosa infection who developed resistance to CAZ-AVI. The authors further went on demonstrating in vitro results of a 7-day HFIM testing the same bacterial strain. Overall, the study is interesting and informative to dosing regimen optimization. However, several points would benefit from a more detailed analysis and better data interpretation.

  1. Though a correlation between in vivo and in vitro studies was observed, it’s not sufficient to come up with the conclusion proposed in Line 41-43. Unless the drug effect on a larger patient population with specified clinical complications (renal impairment, obese, etc.) are investigated and a comprehensive in vitro-in vivo extrapolation are developed and validated, the conclusion should be rephased with cautious.

Thank you for your comment. We have rephrased the conclusion. 

  1. Considering the potential nonlinear synergetic effect between CAZ and AVI, using CAZ plasma concentration as a representative could be controversial. The authors should provide convincing evidence to back up their assumption. Additionally, a dose-response curve is always preferred to better characterize drug effect and further guide dose selection.

We agree with the reviewer that it could be more appropriated to evaluated plasma concentrations of both ceftazidime and avibactam. Indeed, we had considered this as a study limitation. We have emphasized this point in the study limitations. Conversely, based on previous literature and from a clinical point of view, CAZ and AVI displayed a similar PK/PD profile in terms of lung penetration, volume of distribution, time-dependent activity, low plasma protein binding and renal clearance [ref 6,15,23].

  1. A preliminary WGS analysis and supporting data (figure) should be included to demonstrate AmpC is the key mutation involved in CAZ-AVI resistance in this patient. Similarly, HFIM selected drug-resistant strain should be evaluated (verify for AmpC) and discussed to demonstrate it could serve as a qualified in vitro tool for this proof-of-concept study.

As described, WGS analysis was performed in the sequential clinical isolates and in the HFIM mutants. A new table (Table 1) was added to collect this information.

  1. To investigate the CAZ-AVI induced P. aeruginosa resistance, the development of an in vivo animal model is strongly recommended.

Thank you so much for the suggestion. We will take into consideration for further studies.

  1. Data interpretation

(1)    The drug concentrations presented in the graphical abstract and Fig 2 (figure legend) are different from the text, please double check. For HFIM, using 2/4/8X MIC would be better than the corresponding in vivo Css.

Thank you for the comment. Drug concentrations have been double checked in both the graphical abstract and figure 2. Drug concentrations in the case of Figure 2 refer to the resistance studies (A portion of the bacterial suspension was quantitatively cultured onto agar supplemented with CAZ-AVI at two, four, and eight times the reference MIC to evaluate amplification of resistant subpopulations.)

(2)    For Fig 1 and 2, X/Y axis should be labelled properly. A segmented X/Y axis should be considered to enable a clear view of all data points. Also, the number of replicates (n) and error bars (SD/SEM) should be included, statistical significance should be assessed.

Thank you for the comment. Most of proposed changes have been applied and updated in figures.

(3)    To allow a straightforward comparison of dose-dependent resistance, a table would be preferred to sum up results from Fig 2.

Thank you for the suggestion. We have added a table in supplemental material to clarify results (Table Supplementary 1).  

(4)    A continuous timeline for the clinical treatment record (especially for the graphical abstract) will be greatly appreciated.

Done.

(5)    The data from 4 dose groups could be integrated into 1 Bland-Altman plot in supplemental figure.

Thank you for the suggestion. You can find the integrated Bland-Altman plot bellow. However, since we have different targets (Ceftazdime Css of 12,18,30 and 48 mg/L), we think the individual graphics reflect more accurately our data.

(6)    The HPLC method should be included in “Materials and Methods”.

Done. See lines 413-415.

(7)    The cited references should be formatted.

Done.

Reviewer 2 Report

The work aims to correlate observations of a clinical case with in vitro HFIM results, especially focused on the emergence of resistance, being a promising example of translational research, however, multiple improvements could be performed in the presented document. While the document is generally well written, explained and supported, additionally to minor improvements, two major points must be addressed.

Minor improvements:

1 - Line 133 and 134: dots instead of commas in concentration values

2 - Line 193: “(PDC database www.arpbigidisba.com)” should be cited in accordance with the own website “for citation” section.

3 - Line 207: the text is incomplete in the section “…a critically patient with relevant…”

4 - Line 217: instead of “…and 774 mg/l…” should be 77.4 mg/L, according to line 125

5 - Units for concentration should be mg/L and not mg/l, lines: 125, 133, 134, 217, 221, 256.

Major modifications:

1 - Figure 1 and Figure 2 require several improvements:

i) no title and/or units in both axis

ii) According to methodology, the results presented are, at least, the mean of the values obtained in the duplicate assay (number of samples obtained for CFU determination per time-point should be discriminated), hence the graphical representation of results should contain error bars.

iii) The results presented are not in accordance with the values referred in the text previous to the graphical representation of results, namely line 157, where the initial inoculum is designated as 4.16 and 4.48 log10 CFU/mL.

2 – Since no additional information is available to clarify which initial inoculum is correct, if the graphical representation is ignored and we consider only the data provided in text, an initial inoculum of 4.16 and 4.48 log10 CFU/mL would be a relevant information when comparing the selection of resistant mutants, since the mutational frequency is generally much higher. Additionally, these conditions would no longer represent a high inoculum infection, hence the comparisons performed should take this into account as one limitation.

On the other hand, if the correct data is presented in figures 1 and 2, additionally to the improvements suggested, one must observe that none of the effects observed can be considered bactericidal by the designated cut-off value (“Bactericidal activity was defined as a ≥3 log10 CFU/mL reduction at 24 h relative to the initial inoculum”) and this should be taken into account in the discussion and conclusions.

Author Response

The work aims to correlate observations of a clinical case with in vitro HFIM results, especially focused on the emergence of resistance, being a promising example of translational research, however, multiple improvements could be performed in the presented document. While the document is generally well written, explained and supported, additionally to minor improvements, two major points must be addressed.

Minor improvements:

1 - Line 133 and 134: dots instead of commas in concentration values

Done.

2 - Line 193: “(PDC database www.arpbigidisba.com)” should be cited in accordance with the own website “for citation” section.

Done.

3 - Line 207: the text is incomplete in the section “…a critically patient with relevant…”

Thank you for the comment. The right paragraph is: We evaluated a critically patient with relevant comorbidities and ventilator-associated pneumonia and tracheobronchitis due to XDR P. aeruginosa ST175 with a borderline CAZ-AVI MIC at baseline, who developed resistance to CAZ-AVI during treatment. This information has been updated.

4 - Line 217: instead of “…and 774 mg/l…” should be 77.4 mg/L, according to line 125

Done.

5 - Units for concentration should be mg/L and not mg/l, lines: 125, 133, 134, 217, 221, 256.

Done.

Major modifications:

1 - Figure 1 and Figure 2 require several improvements:

  1. no title and/or units in both axis

Done.

  1. According to methodology, the results presented are, at least, the mean of the values obtained in the duplicate assay (number of samples obtained for CFU determination per time-point should be discriminated), hence the graphical representation of results should contain error bars.

Done.

iii) The results presented are not in accordance with the values referred in the text previous to the graphical representation of results, namely line 157, where the initial inoculum is designated as 4.16 and 4.48 log10 CFU/mL.

The data 4.16 and 4.48 log10 CFU/mL does not refer to the initial inoculum, but to the drop that occurs from the inoculum at the end of the experiment. The bacterial density of the starting inoculum was 7.08 log10 CFU/mL, as it can be observed in figures. CAZ-AVI Css of 30 mg/L and 48 mg/L achieved a continuous bacterial reduction of 4.16 and 4.48 log10 CFU/mL, respectively, with final bacterial densities of 2.92 and 2.60 log10 CFU/mL, respectively. We have rephrased this paragraph in the manuscript in order to clarify.

2 – Since no additional information is available to clarify which initial inoculum is correct, if the graphical representation is ignored and we consider only the data provided in text, an initial inoculum of 4.16 and 4.48 log10 CFU/mL would be a relevant information when comparing the selection of resistant mutants, since the mutational frequency is generally much higher. Additionally, these conditions would no longer represent a high inoculum infection, hence the comparisons performed should take this into account as one limitation.

As we have clarified above, the bacterial density of the starting inoculum was 7.08 log10 CFU/mL and the data 4.16 and 4.48 log10 CFU/mL refer to the drop that occurs from the inoculum at the end of the experiment. This initial inoculum represents a high inoculum infection.

On the other hand, if the correct data is presented in figures 1 and 2, additionally to the improvements suggested, one must observe that none of the effects observed can be considered bactericidal by the designated cut-off value (“Bactericidal activity was defined as a ≥3 log10 CFU/mL reduction at 24 h relative to the initial inoculum”) and this should be taken into account in the discussion and conclusions.

We have really appreciated this comment. Although the included definition referred to 24 hours, in this HFIM bactericidal activity was considered as a reduction of 3 log10 CFU/mL from the initial bacterial density (see previous studies of our group: i.e., Montero et al. Microbiol Spectr. 2022. doi: 10.1128/spectrum.00892-22). In line with this, in Css30: 7.08 - 2.92 = 4.16 log10 CFU/mL and in case of Css48: 7.08 - 2.06 = 4.48 log10 CFU/mL. Both shows a ≥3 log10 CFU/mL reduction relative to the initial inoculum. Thus, they are bactericidal.

Reviewer 3 Report

The present manuscript entitled "Suboptimal concentrations of Ceftazidime/Avibactam (CAZ-AVI) may select for CAZ-AVI resistance in extensively drug-resistant Pseudomonas aeruginosa: In vivo and in vitro evidence" by Inmaculada López-Montesinos, Maria Milagro Montero, Sandra Domene-Ochoa, Carla López-Causapé, Daniel Echeverria-Esnal, Lluïsa Sorlí, Nuria Campillo, Sonia Luque, Eduardo Padilla, Nuria Prim, Santiago Grau, Antonio Oliver, Juan P. Horcajada (antibiotics-1968782) is written correctly and has a good structure; moreover, it has all the necessary parts. The article is interesting from a medical point of view; therefore, it should interest the reader. I proposed improvements in the method description and with a presentation of figures. The paper meets Antibiotics' requirements, and I recommend the article for publication in Antibiotics following the common editing stage. My current decision is a minor revision. More specific comments and observations are presented below.

1. Please check if all abbreviations used are previously explained in the text.

2. Please standardize units. Sometimes they are written at the value, sometimes separately. The liter is written once as "l" and once as "L".

3. Page 4, line 134. “91.,5” ?

4. Figure 1. Quality needs to be improved. Please add axes with markers, axis descriptions, and units. Fonts should be standardized. Line and dot thickness should be enhanced to improve visibility.

5. Figure 2. The same like for Figure 1. You can consider removing the titles but numbering them as a), b).

6. The text includes websites that should be mentioned as references, while the References section should also contain access dates.

7. Page 8, line 199. The  “relationship” is mentioned. This term should be changed to "relation". The relationship tends to be used more broadly to describe the interactions between specific people or smaller groups of people.

8. References should appear in ascending order. Please renumber and check other items.

9. Please add the conclusion section.

10. Page 12, line 409. Please add more details about the HPLC method.

11. Please check whether References comply with the journal's requirements.

12. What are the plans for the future?

I hope that the comments presented will help improve the article.

Author Response

The present manuscript entitled "Suboptimal concentrations of Ceftazidime/Avibactam (CAZ-AVI) may select for CAZ-AVI resistance in extensively drug-resistant Pseudomonas aeruginosa: In vivo and in vitro evidence" by Inmaculada López-Montesinos, Maria Milagro Montero, Sandra Domene-Ochoa, Carla López-Causapé, Daniel Echeverria-Esnal, Lluïsa Sorlí, Nuria Campillo, Sonia Luque, Eduardo Padilla, Nuria Prim, Santiago Grau, Antonio Oliver, Juan P. Horcajada (antibiotics-1968782) is written correctly and has a good structure; moreover, it has all the necessary parts. The article is interesting from a medical point of view; therefore, it should interest the reader. I proposed improvements in the method description and with a presentation of figures. The paper meets Antibiotics' requirements, and I recommend the article for publication in Antibiotics following the common editing stage. My current decision is a minor revision. More specific comments and observations are presented below.

  1. Please check if all abbreviations used are previously explained in the text.

Done.

  1. Please standardize units. Sometimes they are written at the value, sometimes separately. The liter is written once as "l" and once as "L".

Done.

  1. Page 4, line 134. “91.,5” ?

Done.

  1. Figure 1. Quality needs to be improved. Please add axes with markers, axis descriptions, and units. Fonts should be standardized. Line and dot thickness should be enhanced to improve visibility.

Done.

  1. Figure 2. The same like for Figure 1. You can consider removing the titles but numbering them as a), b).

Done.

  1. The text includes websites that should be mentioned as references, while the References section should also contain access dates.

Done.

  1. Page 8, line 199. The  “relationship” is mentioned. This term should be changed to "relation". The relationship tends to be used more broadly to describe the interactions between specific people or smaller groups of people.

Done.

  1. References should appear in ascending order. Please renumber and check other items.

Done.

  1. Please add the conclusion section.

Done.

  1. Page 12, line 409. Please add more details about the HPLC method.

Done. See lines 413-415.

  1. Please check whether References comply with the journal's requirements.

Done

  1. What are the plans for the future?

Clinical and microbiological studies with a major collection of isolates are needed to assess the feasibility, effectiveness, and safety of this translational research. Further in vivo and in vitro examples should be analysed before drawing generalizable conclusions.

Round 2

Reviewer 1 Report

I would like to congratulate the authors for the revision work and thank them for answering and clarifying almost all the aspectsof my questions and comments. Meanwhile, a few points could be conducted to strengthen the study:

1. Add the number of replicates applieded in ex vivo culture;

2. Use professional softwares like Prism GraphPad to generate the figures;

3. Unless different methods were adopted for the ceftazidime concentration prediction, the Bland-Altman plot should be integrated into 1 figure using the same SD (doi.org/10.11613/BM.2015.015).

Author Response

Reviewer 1

I would like to congratulate the authors for the revision work and thank them for answering and clarifying almost all the aspects of my questions and comments. Meanwhile, a few points could be conducted to strengthen the study:

  1. Add the number of replicates applied in ex vivo culture.

HFIM was performed in duplicate. This information has been emphasized in methods and in the foot of figures 1 and 2. In order to improve the appearance of figure 1 and 2, data obtained of HFIM assays have been detailed in Table Supplementary 1.

  1. Use professional softwares like Prism GraphPad to generate the figures.

Done. Please, note that if the error bar is short than the size of the symbol, Prism GraphPad will not draw it. SD values ​​can be found in the table of the supplementary material. Figures have been made again with the GraphPad program.

  1. Unless different methods were adopted for the ceftazidime concentration prediction, the Bland-Altman plot should be integrated into 1 figure using the same SD (doi.org/10.11613/BM.2015.015).

Thank you for the comment. We have included the integrated Bland-Altman plot in the supplementary material.

Reviewer 2 Report

I wish to thank the authors for addressing and clarifying most of the comments and congratulate on the revision, this document is more complete and clear for the reader. With some minor revisions I believe the final document will be ready for publication.

1) Lines 158-159: "respectively" used twice in quick succession

2) line 219: "critically patient" where I believe the intended text should read "critically ill patient"

3) The appearance of figures 1 and 2 could be improved (for example by removing the excess of lines in the background or diminishing the size of markers and/or lines)

4) The standard deviation values should be reported whenever suitable (ex: figure 2 and in Table Supplementary 1)

Author Response

Reviewer 2

I wish to thank the authors for addressing and clarifying most of the comments and congratulate on the revision, this document is more complete and clear for the reader. With some minor revisions I believe the final document will be ready for publication.

  • Lines 158-159: "respectively" used twice in quick succession

Done.

  • line 219: "critically patient" where I believe the intended text should read "critically ill patient"

Done.

  • The appearance of figures 1 and 2 could be improved (for example by removing the excess of lines in the background or diminishing the size of markers and/or lines)

Thank you for the suggestion. Figures 1 and 2 have been improved. Prism GraphPad has been applied to generate the figures (See also reviewer 1 comment 2).

  • The standard deviation values should be reported whenever suitable (ex: figure 2 and in Table Supplementary 1)

The standard deviation values have been added in Table Supplementary 1 (See also reviewer 1 comment 1).